# Smectite as a Preventive Oral Treatment to Reduce Clinical Symptoms of DSS Induced Colitis in Balb/c Mice

**DOI:** 10.3390/ijms22168699

**Published:** 2021-08-13

**Authors:** Anne Breitrück, Markus Weigel, Jacqueline Hofrichter, Kai Sempert, Claus Kerkhoff, Nooshin Mohebali, Steffen Mitzner, Torsten Hain, Bernd Kreikemeyer

**Affiliations:** 1Extracorporeal Immunomodulation Unit (EXIM), Fraunhofer Institute for Cell Therapy and Immunology (IZI), 18057 Rostock, Germany; jacqueline.hofrichter@izi.fraunhofer.de (J.H.); steffen.mitzner@izi.fraunhofer.de (S.M.); 2Division of Nephrology, Department of Internal Medicine, University Medicine Rostock, 18057 Rostock, Germany; 3Institute of Medical Microbiology, Justus Liebig University, 35392 Giessen, Germany; markus.weigel@mikrobio.med.uni-giessen.de; 4Queensland Brain Institute, The University of Queensland, 4072 St Lucia, Brisbane 4000, Australia; kai.sempert@uq.edu.au; 5Department of Human Sciences, School of Human Sciences, University of Osnabrück, 49076 Osnabrück, Germany; claus.kerkhoff@uni-rostock.de; 6Institute of Medical Microbiology, Virology and Hygiene, University Medicine Rostock, 18057 Rostock, Germany; nooshin.mohebali@med.uni-rostock.de; 7German Center for Infection Research (DZIF), Partner Site Giessen-Marburg-Langen, 35392 Giessen, Germany

**Keywords:** IBD, DSS-colitis, diosmectite, clay-mineral, microbiota

## Abstract

Natural smectites have demonstrated efficacy in the treatment of diarrhea. The present study evaluated the prophylactic effect of a diosmectite (FI5pp) on the clinical course, colon damage, expression of tight junction (TJ) proteins and the composition of the gut microbiota in dextran sulfate sodium (DSS) colitis. Diosmectite was administered daily to Balb/c mice from day 1 to 7 by oral gavage, followed by induction of acute DSS-colitis from day 8 to 14 (“Control”, n = 6; “DSS”, n = 10; “FI5pp + DSS”, n = 11). Mice were sacrificed on day 21. Clinical symptoms (body weight, stool consistency and occult blood) were checked daily after colitis induction. Colon tissue was collected for histological damage scoring and quantification of tight junction protein expression. Stool samples were collected for microbiome analysis. Our study revealed prophylactic diosmectite treatment attenuated the severity of DSS colitis, which was apparent by significantly reduced weight loss (*p* = 0.022 vs. DSS), disease activity index (*p* = 0.0025 vs. DSS) and histological damage score (*p* = 0.023 vs. DSS). No significant effects were obtained for the expression of TJ proteins (claudin-2 and claudin-3) after diosmectite treatment. Characterization of the microbial composition by 16S amplicon NGS showed that diosmectite treatment modified the DSS-associated dysbiosis. Thus, diosmectites are promising candidates for therapeutic approaches to target intestinal inflammation and to identify possible underlying mechanisms of diosmectites in further studies.

## 1. Introduction

Inflammatory bowel disease (IBD) is a chronic and relapsing inflammatory disease of the intestine with severe, occasionally bloody diarrhea with high-frequency abdominal pain, weight loss, anemia, fever and general physical weakness [1]. Severe outcomes include toxic megacolon, abscesses, colorectal cancer, fistulas and extra-intestinal manifestations [2]. Ulcerative colitis (UC) and Crohn’s disease (CD) are the two most common entities of IBD, differing in the involved gut regions, the type of inflammation and the applied therapeutic procedures [3]. The cause of IBD is multifactorial and has a complex pathogenesis which is still not completely resolved, including host and environmental factors that contribute to the disease and its severity [4]. The current hypothesis is that, in genetically predisposed individuals, an inappropriate immune response of the gastrointestinal immune system against the luminal microbiome is triggered by environmental factors, which results in the onset or reactivation of IBD. Current IBD therapies, e.g., aminosalicylates, corticosteroids, immunosuppressive agents and biologics, focus on the achievement and maintenance of clinical, biochemical, endoscopic and histological remission [5,6]. However, currently no cure for IBD exists and patients require life-long medication. For these reasons, an urgent need to identify new therapeutic strategies beside the conventional medicine is eminent, as this could improve treatment options and the life quality of IBD patients.

Natural clay minerals and their healing properties were already mentioned by Aristotle (384–322 BC) and their external and internal use has been documented over several centuries [7]. Clay minerals consist of different layers of silicates and have been widely used for the treatment of gastrointestinal diseases such as diarrhea and irritable bowel syndrome (IBS) [8,9,10]. Therapeutic administration of the natural silicate diosmectite in hapten trinitrobenzene sulphonic acid (TNBS)-induced colitis reduced related morphological signs, biochemical markers and decreased the severity of diarrhea in rats [11]. Furthermore, the therapeutic benefits were comparable to standard medication with sulphasalazine. In clinical studies including patients with mild-to-moderate UC, a combined diosmectite and mesalazine treatment induced significantly higher clinical remission and mucosal healing rates compared to a placebo group as shown in a randomized, placebo-controlled study [12]. We previously studied the therapeutic potential of a specific natural silicate from Friedland (Germany) on iodoacetamide-induced inflammation in a defined bowel segment of rats. Particularly, histological examinations of intestine, cytokine expression and cell infiltration into intestine segments in a short-term inflammation model revealed beneficial effects of the administered smectite [13]. However, since the mineral was applied therapeutically and within a short-term inflammation model the effects of this smectite in a long-term, well-establish experimental colitis model are still unknown.

For the current study we hypothesized that a prophylactic smectite treatment attenuates the clinical course of dextran sulphate sodium (DSS) colitis. Herein, diosmectite FI5pp from Friedland (Germany) was administered before induction of acute DSS colitis in Balb/c mice. Besides clinical parameters and histological alteration of colonic tissue, the effect of diosmectite on the expression of tight junction (TJ) proteins within the colon as well as the gut microbiome, was investigated. 

## 2. Results

### 2.1. Prophylactic FI5pp-Treatment Attenuates the Clinical Course of DSS-Induced Colitis

Mice without DSS-induced colitis showed an increase in body weight during the experiment (Figure 1A). In contrast, mice treated with DSS alone exhibited a temporary weight loss of almost 18% until day 15, which was recovered only partially over the course of this experiment. However, the proportional body weight loss in FI5pp + DSS-treated mice was lower over the entire experiment reaching significantly higher weights from day 14 to 15 (*p* = 0.043 and *p* = 0.022 vs. DSS, respectively) (Figure 1A). Monitoring of clinical symptoms confirmed our observation. The disease activity index (DAI) scores showed a continuous increase with the progression of DSS-treatment, whereas FI5pp administration reduced this effect on the DAI from day 12 to 21, although the score was higher than in control mice (Figure 1B). The maximum score in mice treated with DSS alone was obtained on day 15, with 6 points on average, while FI5pp-treated mice had a significantly reduced maximum score of 4 points on average on day 15 (*p* = 0.0025 vs. DSS). 

Histological analysis of the colon in control animals showed a normal morphology of crypts and no signs of inflammation. The colon of DSS mice without treatment revealed multifocal crypt distortion, infiltration of inflammatory cells, extensive epithelial damage (indicated by arrow, arrowhead and red circle in Figure 2A) and complete destruction of the tissue architecture. Contrary, the colons from prophylactic FI5pp-treated DSS mice exhibited only moderate evidence of inflammatory cell infiltration and mucosal injury (Figure 2A). Furthermore, FI5pp significantly reduced the DSS-induced histological score to 8.8 compared to DSS alone treated animals with 15.3, but not to control levels of 3.6 (Figure 2B). However, FI5pp did not prevent the DSS-induced shortening of the colon (Figure 2C).

### 2.2. Prophylactic FI5pp-Treatment Suppresses Reduction of Tight Junction Protein Expression in DSS-Induced Colitis

We studied the expression of selected TJ proteins such as claudin-2 and -3 in proximal and distal colon tissue (Figure 3A,B). The total protein level of claudin-3 in proximal colon was unaltered during DSS-colitis compared to control mice. However, claudin-3 expression in distal colon was significantly decreased by DSS colitis (*p* = 0.031 vs. DSS), which was not apparent after FI5pp treatment. The claudin-2 level in proximal colons was only slightly enhanced after DSS-treatment but did not reach a significant different value. Similar but less pronounced effects were observed for claudin-2 expression in mice treated with FI5pp. 

### 2.3. Prophylactic FI5pp-Treatment Attenuates Bacterial Diversity in Mouse 

A total of 121 different genera were classified for the “Control” (n = 3), “DSS” (n = 6) and “FI5pp + DSS” (n = 5) samples. Of those genera, 93 have a relative abundance of below 1% in all 14 samples across the analyzed groups. *Bacteroidales S27-7 group ge* and *Bacteroides*, two commensals in mouse stool, were the most abundant genera in all three groups. We observed a strong decrease of *Bacteroidales S27-7 group ge* in “DSS” and “FI5pp + DSS” compared to the “Control” group. *Bacteroides* increased in “DSS”, but slightly decreased in “FI5pp + DSS”. *Lachnospiraceae unclassified* decreased in “DSS” and “FI5pp + DSS”, compared to the “Control” group. The *Lachnospiraceae NK4A136* group was at the same level in all three groups. *Prevotellacea UCG-001* was most abundant in the “DSS” group and had the lowest abundance in the “Control” group. *Akkermansia* and *Alistipes* were exclusively present in an extremely low relative abundance in the “Control” group. *Akkermansia* showed the strongest presence in the “DSS” group and *Alistipes* in the “FI5pp + DSS” group (Figure 4A). 

The rarefaction curve based on the number of observed operational taxonomic units (OTU) for the different samples shows no difference among the three groups (Figure 4B). Principle coordinates analysis (PCoA) of Bray-Curtis dissimilarity displayed distinct groups for “Control”, “DSS” and “FI5pp + DSS” samples (Figure 4C). A pairwise analysis of molecular variance (AMOVA) confirmed a significant spatial separation observed between the “Control”, “DSS” and “FI5pp + DSS” groups. Homogeneity of molecular variance (HOMOVA) showed no significantly different amount of variation between the samples of the three groups. 

To verify the changes of the seven genera described above on OTU level, we analyzed the differences between the three groups with Metastats (Appendix A). While all 1.614 OTUs were analyzed, we focused on the 100 OTUs with the highest total abundance, which represented 81.40 % to 99.91 % of all reads mapped to the described genera. *Bacteroidales S27-7group ge* represented with 22 distinct OTUs the most common genus in the 100 most abundant OTUs. Eleven of these showed a significantly different expression across all three groups and primarily confirm the higher abundance of *Bacteroidales S27-7group ge* in “Control” compared to “DSS” and “FI5pp + DSS”. The only significant change for the Bacteroides genus, represented by three OTUs, is the lower amount of reads for OTU0001 in “DSS” than in “FI5pp + DSS”. Of the ten OTUs present, only two showed significant changes for the *Lachnospiraceae NK4A136* group. OTU0041 was increased in “Control” versus. “FI5pp + DSS” and OTU0072 in “Control” versus. “DSS”. Both OTUs represented only a small fraction of the total number of reads and confirm the overall same relative abundance for the *Lachnospiraceae NK4A136* group. *Lachnospiraceae unclassified* genus was distributed over 17 OTUs and 7 of those OTUs showed a significant different abundance. While we saw some evidence for the changes on genus level, the total change is not explained by the 100 OTUs with the highest total abundance alone, but due to the large amount of smaller changes outside the top 100. *Prevotellacea UCG-001* and *Akkermansia* rendered as a single OTU (OTU0002 and OTU0004, respectively). Both were significantly increased in “DSS” compared to “Control” and “FI5pp + DSS”. Additionally, *Akkermansia* was significantly increased in “FI5pp + DSS” versus. “Control”. *Alistipes* (OTU0005) was significantly increased in “FI5pp + DSS” compared to “Control” and “DSS”. A second OTU for the genus *Alistipes* (OTU0015) was increased in “DSS” and “FI5pp + DSS” compared to “Control”.

## 3. Discussion

Depending on their mineralogical structure, natural silicate minerals are characterized by a large surface area and a strong ion-exchange capacity, which allows high adsorption potential of water, toxins and bacteria. Due to these properties, natural silicate minerals have been used for decades in the treatment of diarrheal conditions and reflux diseases [14,15,16]. In addition, an intestinal anti-inflammatory potential could be demonstrated in a few human and animal studies [9,10,11,12]. However, to date and to the best of our knowledge, there has been no study to evaluate the prophylactic potential of diosmectite in intestinal inflammatory conditions. In the present in vivo study, an acute DSS-induced colitis model was selected to determine the effects of a prophylactic application of a diosmectite from Friedland (Germany), named FI5pp on clinical parameters, the expression of tight junction proteins and the composition of the gut microbiota. 

In this in vivo study, we found that the clinical course of DSS-induced colitis, characterized by weight lost and increased DAI, was significantly attenuated by prophylactic treatment with FI5pp compared to the untreated DSS-group. The histological examination of the colon remarkably confirmed these findings. DSS-treated mice showed marked morphological epithelial damages and cell infiltration into colon tissue resulting in a high overall histological score and colon shortening. In contrast, FI5pp treatment prevented DSS-induced morphological changes and significantly reduced the histological score. However, it failed to prevent shortening of the colon. These data correspond with the findings in our previous studies, where the application of a diosmectite also decreased histological alteration and improved the overall histological score [13]. Another animal study from González et al., which used a TNBS colitis model in rats, showed that the diosmectite treatment resulted in amelioration of the intestinal morphological signs and reduced the severity of diarrhea [11]. Similar data was also obtained in a clinical study from Jiang XL et al., which investigated the efficacy of a combined diosmectite and mesalazin treatment for active, mild-to-moderate ulcerative colitis, where a significantly higher clinical remission and endoscopic mucosal healing rate compared to a placebo group was reported [12]. Although our study focused on the prophylactic approach, thus has limited comparability with therapeutic studies, the underlying functional mechanisms of diosmectites may be similar. 

It is well known that IBD is associated with a disturbed intestinal epithelial barrier, which is due to pathologically altered expression of tight junction proteins. As a consequence, an uncontrolled translocation of luminal substances throughout the epithelium results in an overactive mucosal immune response [17]. The destabilizing altered expression of tight junction proteins is highly influenced by the release of pro-inflammatory cytokines, such as tumor necrosis factor-α (TNF-a) and interferon-γ (IFNg), which are released as a consequence of the ongoing intestinal inflammation processes [18]. An upregulation of pore-forming claudin-2 and downregulation of sealing claudin-3 has been reported in ulcerative colitis, as well as in experimental colitis models, leading to altered TJ structure and barrier dysfunction [19,20,21,22]. In our study, we observed a significantly decreased claudin-3 expression in distal colon of DSS alone-treated animals, whereas no changes in claudin-3 expression could be observed in the proximal colon. As the highest claudin-2 expression has been reported in the small intestine and the ileocecal junction [23,24], we investigated the proximal colon only. The claudin-2 expression in proximal colons was slightly but not significantly upregulated in DSS-treated animals. As a positive correlation between active disease state and claudin-2 abundance has been reported, our experimental protocol, which included additional 7 days of recovery after DSS application, might provide an explanation for the missing significant effects [21,25,26]. Thus, acute intestinal inflammatory processes may have subsided. The prophylactic application of FI5pp resulted in a slightly but not significantly enhanced claudin-3 expression, whereas claudin-2 was not affected. In contrast to our study, an in vivo feeding study with early weaned pigs showed that smectite application modulates the tight junction protein profile. The expression of other stabilizing TJ proteins, namely, occludin, claudin-1 and zonula occludens-1 were increased, whereas inflammatory cytokine expression was decreased [27]. Another in vitro study by Mahraoui et al. also demonstrated a barrier-strengthening effect of a smectite. Incubation of the intestinal epithelial cell line HT29 with smectite prevented a TNF-α and INF-γ-induced flux of 14C-mannitol and horseradish peroxidase in double chamber cell culture experiments [28]. Compared to our study, the observed effects on the intestinal epithelial barrier and the cytokine milieu were linked to therapeutic smectite applications. A modified experimental protocol in regard to application duration, frequency and smectite concentration may result in significant changes of TJ protein expression. 

In mammals, the gut microbiome is associated with physiology, human developmental processes and has an important role in human health and disease status [29,30]. There are numerous studies providing evidence of gut microbial influences on immunity, metabolism, neurodegenerative processes and even human psychology [31,32,33,34]. Therefore, the gut microbiome was studied in all treatment groups at the endpoint of the experiment. We found several changes in gut microbial composition concerning the *Akkermansia* genus, *Alistipes* spp., *Bacteroidales S24-7, Lachnospiraceae NK4A136* group and *Prevotellaceae UCG-001.* The *Akkermansia* genus is part of the *Verrucomicrobia* phylum and has currently two members (*A. glycaniphila* and *A. muciniphila*), which are both strictly anaerobic mucin-degrading bacteria [35,36]. *A. glycaniphila* was isolated from python faeces while *A. muciniphila* is commonly reported in mouse feces and gut. Multiple studies associated an increased *Akkermansia* population with a lower disease severity [37,38], while other reports describe *A. muciniphila* as a pathobiont which promotes inflammation and colitis [39]. The increased *Akkermansia* population in our DSS group suggests the latter case. The nearly complete absence of *Alistipes* spp. in the control samples seems atypical and remains unexplained. A reduction of *Alistipes* in DSS group compared to FI5pp + DSS treated mice was also observed in patients with IBD, UC and CD [40]. Furthermore, it was shown that *Alistipes finegoldi* attenuated DSS-induced colitis in mice [41]. *Bacteroidales S24-7* is a family of mostly uncultured bacteria which are dominant in the mouse gut and other mammals [42,43]. Consistently, *Bacteroidales S24-7* has the highest overall abundance in our data for all three groups and was considerably reduced in the two DSS groups. Multiple *Bacteroides spp.* have been reported to induce or enhance colitis in mice [41,44], but to date it remains unclear how exactly *Bacteroidales S24-7* contributes to the gut homeostasis in mice.

The classification of the OTUs within the *Lachnospiraceae NK4A136* group and *Prevotellaceae UCG-001* leads, in most cases, to uncultured bacteria in the SILVA ribosomal RNA gene database, which restricts conclusions for the genus and function of these OTUs in the mouse gut. The *Lachnospiraceae* family, represented by the *Lachnospiraceae unclassified* and *Lachnospiraceae NK4A136* group, is a strong producer of butyrate and other short-chain fatty acids [45] which are absorbed by the colon and utilized in the metabolism of colonocytes [46]. A reduction in Lachnospiraceae was shown to be involved in the formation of visceral hypersensitivity, a characteristic symptom of IBD, in rats [47]. We conclude that the observed increase of *Lachnospiraceae* in FI5pp + DSS treated mice, compared to the DSS group, is an indication for a normalization of the gut microbiome to a level shown in animals of the control group. 

While some studies report a decrease in *Prevotella* spp. in fecal and colon samples from mice with IBD [48,49], others report an increase in *Prevotella* in the mucosa-associated flora and in fecal samples from human patients with UC or IBD [50,51]. Similarly, the presence of *Prevotella* spp. has been negatively [52] and positively [41,53] associated with severity of IBD in mice. Furthermore, Iljazovic et al. showed that a *Prevotella intestinalis*-induced decrease in interleukin 18 exacerbates colonic inflammation [53]. Hence, the small decrease in *Prevotellaceae UCG-001* in the FI5pp + DSS group compared to the DSS group in our microbiome analysis could be an explanation for the weaker inflammatory response of the colon.

Overall, our study shows that prophylactic diosmectite treatment has beneficial effects on ulcerative colitis development in mice. We found positive effects on the clinical symptoms associated with DSS-colitis, such as weight loss, stool consistency and occult blood. Furthermore, the positive impact of diosmectite on the histological score, as observed in a previous in vivo study, could be confirmed [13]. Analysis of the gut microbiota composition identified a possible microbiome shaping potential. In contrast, only minor effects on the TJ expression could be demonstrated. For the first time, we demonstrated the prophylactic benefits of diosmectite on the end phase of the acute inflammatory processes in a DSS-colitis model. Further in vivo studies should consider the modification of the experimental protocol in terms of application duration, frequency and concentration of the diosmectite. Especially the treatment of healthy mice with the diosmectite (FI5pp-treated group alone) would provide general information about the effect on weight and stool consistency and would allow insights into possible underlying mechanisms. Furthermore, it would be of interest to use the diosmectite in a chronic DSS colitis model to see if the prophylactic treatment in a model with multiple cycles of DSS could ameliorate or even prevent mice from relapse. Moreover, the investigation of therapeutic effects in this model is also of interest and is currently under investigation. The exploration of the underlying mechanism that could explain the in vivo efficacy and other favoring effects of diosmectites on colitis-related diseases could be the focus of future studies. 

## 4. Materials and Methods

### 4.1. Animal

Male Balb/c mice obtained from Charles River Laboratories (Sulzfeld, Germany) were grouped housed with 3–8 mice per cage, under controlled environmental conditions regarding temperature and humidity, with a 12:12 h light:dark cycle. They received standard food and water *ad libitum*. Mice aged 15 weeks (average body weight 26 g) were randomly assigned to different experimental groups. All animal experiments were performed in compliance with the German animal protection law and were approved by the local animal care and use committee (7221.3-1.1-029/13).

### 4.2. Animal Treatment Protocol 

Prophylactic treatment (Figure 5) was initiated one week prior to colitis induction (days 1–7). Mice were treated daily with FI5pp orally by gavage (0.5 g/kg BW) in a final volume of 125–150 µL (depending on weight) for 7 days (“FI5pp + DSS”, n = 11). FI5pp was suspended in autoclaved water (10% *w*/*v*). Healthy control animals (“control”, n = 6) and DSS-treated animals (“DSS”, n = 11) received a solution of saline (NaCl, 0.9%) in a final volume of 125-150 µL (depending on weight). Acute colitis was induced for 7 days (days 8–14) by 4.5% DSS (MW 36-50kDA, MP Biomedicals, Eschwege, Germany) dissolved in drinking water. Fresh DSS solution was prepared every second day. Mice were euthanized 7 days after the last DSS application (day 21) by an overdose of intraperitoneally injected ketamine-xylazine (90/25 mg/kg BW). The whole colon was removed and the length was measured. Small sections of the proximal and distal colon were collected and stored in liquid nitrogen for protein isolation. The remaining colon was opened along its long axis, unrolled and was fixed in a 10% formalin solution and embedded in paraffin for hematoxylin and eosin (H&E) staining. 

The experiment was repeated twice (experiment 1 and experiment 2) with 3–8 mice per group, in total 6 “control” mice, 14 “DSS” mice and 11 “FI5pp + DSS” mice. Four mice from the DSS group had to be euthanized prematurely because they reached the predetermined humane endpoint criteria and had to be excluded from the final evaluation. This resulted in a total number of 10 mice for the DSS group. Data from separate experiments (Expt. 1, Expt. 2) were combined for the evaluation. 

### 4.3. Clinical Evaluation of Colitis

Changes in body weight, stool texture/consistency and fecal blood were scored every day to determine the DAI. The DAI was determined according to Cooper et al. with minor modifications, as detailed in Table 1 [54]. was calculated by dividing the body weight on the specified day by the body weight on the first day of DSS treatment (day 8). Feces were screened for blood using a HemoCARE occult blood detection kit (Care Diagnostica, Voerde, Germany). 

### 4.4. Colon Histology

After fixing and embedding the tissue in paraffin, thin sections (5 µm) were cut and stained with H&E using standard methods. Pathological changes in colon sections were evaluated using a modified pathology score for evidence of inflammatory damages as described previously [54] and as detailed in Table 2. Tissue sections were scored separately for score inflammation (inflammation x percent involvement), score extent (extend x percent involvement) and score crypt damage (crypt damage x regeneration). The total histological score was generated by adding the single scores to a maximum value of 36. Each section of colon was analyzed blinded to the experimental group. 

### 4.5. Western Blot Analysis

Colon proteins were extracted using RIPA buffer supplemented with protease inhibitors (Roche, Basel, Switzerland). Briefly, frozen proximal and distal samples were powdered using a homogenizer PeqLab precellys24 (PeqLab Biotechnology, Erlangen, Germany). The homogenized samples were centrifuged at 10,000× *g* for 5 min at 4 °C and the supernatant was collected. Total protein concentrations were determined using a BCA Protein Assay Kit (Pierc, Rockford, IL). A total of 25 µg protein/sample was used with three biological replicates each. Proteins were separated by 12% sodium dodecyl sulfate polyacrylamide gel electrophoresis (SDS-Page) and the gels were then transferred onto a nitrocellulose filter membrane or polyvinylidene difluoride (PVDF) (BioRad, Hercules, CA, USA). Membranes were subsequently blocked in 5% skim milk and incubated (overnight at 4 °C) with primary antibodies against claudin-2 (1:500; Invitrogen, Waltham, MA, USA), claudin-3 (1:500; AAT Bioquest, Sunnyvale, CA, USA) and ß-actin (1:10.000; Sigma-Aldrich, St. Louis, MO, USA). The membranes were subsequently washed and incubated (90 min at room temperature) with horseradish peroxidase-conjugated secondary antibodies. The immunoblots were developed with the Western Blot Luminescence Reagent and exposed by a gel documentation system (Fusion FX, Vilber Lourmat; Eberhardzell, Germany). Densiometric analyses were performed using Fusion-Capt. ß-actin was used as loading control for normalization.

### 4.6. Microbial DNA Extraction

In order to identify differential influences of the treatments on the gut microbiome at the endpoint of the experiment, the bacterial diversity in the mice stool samples was assessed. In experiment 2, stool samples were collected from mice on day 21 (“Control”, n = 3; “DSS” n = 6; “DSS + FI5pp”, n = 5). Bacterial genomic DNA from fecal sample (~35 g) was extracted using a QUIAamp DNA Stool Mini Kit (Quiagen, Hilden, Germany) according to the manufacturer’s instructions.

### 4.7. Library Preparation, 16S Sequencing Run and Data Analysis

The isolated DNA was amplified with the bacterial 16S ribosomal RNA (rRNA) gene primers Bakt_341F (CCTACGGGNGGCWGCAG) and Bakt_805R (GACTACHVGGGTATCTAATCC). The PCR amplicon and the PCR index, a quantity and a quality control, and the sequencing of the individual libraries as a pool in one Illumina MiSeq run were performed as described in the Illumina “16S Metagenomic Sequencing Library Preparation” protocol (www.illumina.com).

Sequencing on the MiSeq platform using the MiSeq Reagent Kit v2 resulted in between 115.022 and 225.305 paired end reads with the length of 250 nt. The microbiome analysis was executed using Mothur (Version 1.43.0, Ann Arbor, MI, USA) [55]. Paired end reads were joined and primer regions removed and filtered for the expected amplicon length of 253 nt ± 10 nt, excluding sequences which contained ambiguous nucleotides. Joined paired end reads were aligned to the SILVA ribosomal RNA gene database trimmed to contain only the hypervariable region V4 and clustered with a similarity threshold of 97% [56]. After chimera removal using VSEARCH, a total of 1.989 OTUs representing between 67.618 and 135.082 paired end reads were obtained and classified against the SILVA ribosomal RNA gene database [57]. For further analyses we subsampled all samples to 60.000 reads, resulting in 1.614 OTUs. Rarefaction curves, PCoA of the Bray-Curtis dissimilarity, AMOVA, HOMOVA and Metastats were created/executed using Mothur [58]. Tables and graphs were created using Microsoft Excel (Version 2017, Redmond, WA, USA). Results with a *p*-value > 0.05 were considered significant.

### 4.8. Statistical Analysis

All data were statistically analyzed using GraphPad Prism 6 (GraphPad Software Inc., San Diego, CA, USA). Data are expressed as mean ± SEM or mean ± SD. Data from separate experiments (Expt. 1, Expt. 2) were combined for statistical analyses and presentation in figures and tables. Group and time effects of clinical parameters (weight loss and DAI) were analyzed by two-way ANOVA for repeated measurements. To analyze the impact of DSS-colitis, we first compared “control” and “DSS” groups. To analyze the influence of the treatment during course of colitis, comparison was made between “DSS” and “FI5pp + DSS”. Histological score and protein expressions were analyzed by Mann-Whitney U test. *p* < 0.05 was considered the criterion for statistical significance. A probability level of *p* < 0.01 was considered very significant.

### 4.9. Data Availability

Microbiome sequencing data have been submitted to the NCBI Short Read Archive repository under the BioProject accession number PRJNA694316 (https://www.ncbi.nlm.nih.gov/sra/PRJNA694316) last accessed date 6 August 2021.

## Figures and Tables

**Figure 1 ijms-22-08699-f001:**
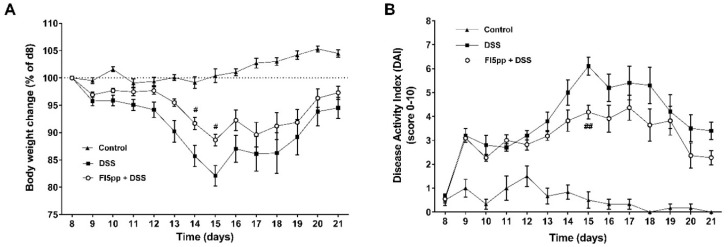
Severity of DSS-induced colitis was determined by proportional body weight change (**A**) and disease activity index (DAI) (**B**). Results are expressed as the mean ± SEM (control n = 6, DSS n = 10, FI5pp+DSS n = 11). Statistical analysis was performed by two-way ANOVA for multiple comparison. #, *p* < 0.05 and ##, *p* < 0.01, vs. DSS mice.

**Figure 2 ijms-22-08699-f002:**
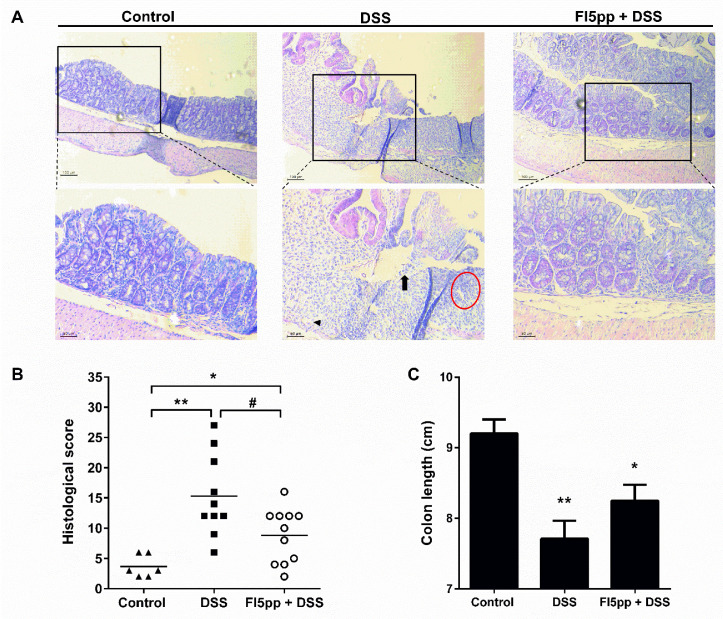
Histological changes in the colon are presented in representative histology images of hematoxylin and eosin (HE) stained colon sections (**A**). HE staining showed crypt distortion (arrow), epithelial damage (arrowhead) and inflammatory cell infiltration (red circle) in colon sections from DSS-treated mice. Corresponding histological score (**B**) and colon length (**C**) of mice with or without prophylactic FI5pp treatment. Results are expressed as mean ± SEM (“Control”, n = 6; “DSS”, n = 10; “FI5pp+DSS”, n = 11). *, *p* < 0.05, vs. control mice; **, *p* < 0.01 vs. control mice, #, *p* < 0.05, vs. DSS mice, as determined by Mann-Whitney U test. Scale bars: 100 µm (upper image panel) and 50 µm (lower image panel).

**Figure 3 ijms-22-08699-f003:**
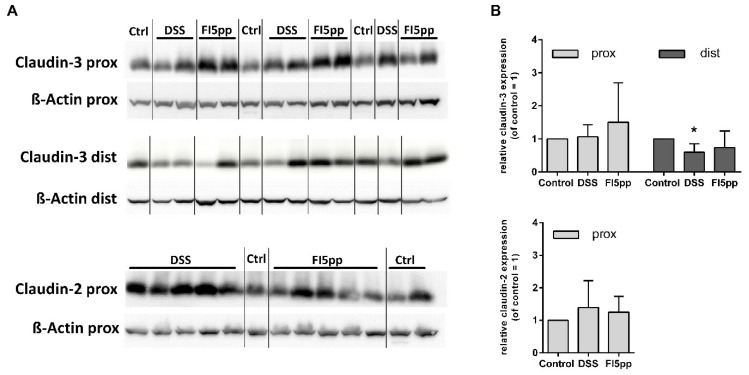
Representative Western blot images of claudin-2 protein (in proximal colonic tissue) and claudin-3 protein (in proximal and distal colonic tissues). Each lane represents an individual mouse (**A**). Quantification of relative expression levels of tight junction proteins claudin-2 and claudin-3 compared to control (**B**). Protein expression was quantified using densitometry analysis. Specific protein expression was normalized to β-actin expression. The results are shown as means ± SD (n = 6). *, *p <* 0.05, vs. control group, as determined by Mann-Whitney U test.

**Figure 4 ijms-22-08699-f004:**
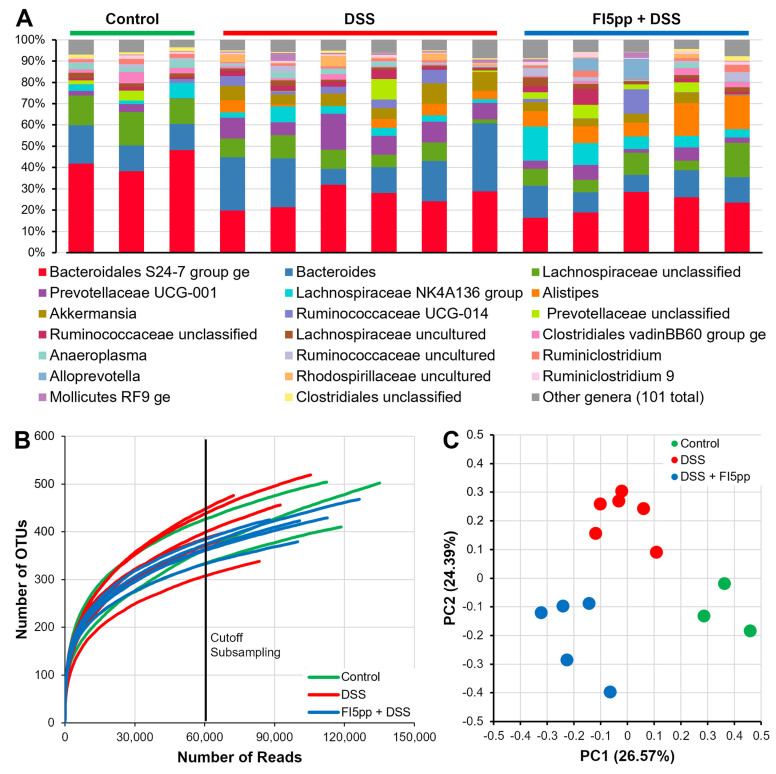
Comparison of the murine fecal microbiome of “Control”, “DSS” and “FI5pp + DSS” groups. Cumulative bar charts of the relative abundance for the top 20 genera found in the 14 mice stool samples. Genera not in the top 20 by relative abundance are categorized as other genera (101 in total) (**A**). Rarefaction curve based on the number of observed OTUs in the 14 samples. Black line indicates the subsampling cutoff used (**B**). PCoA of Bray-Curtis dissimilarity for “Control”, “DSS” and “FI5pp + DSS” groups (**C**).

**Figure 5 ijms-22-08699-f005:**
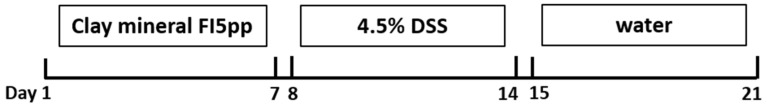
Animal treatment protocol for prophylactic treatment and induction of acute colitis.

**Table 1 ijms-22-08699-t001:** Disease activity index (DAI) score parameters.

Score	Stool Consistency	Bleeding	WEIGHT LOSS
0	Formed	No bleeding	No weight loss
1	Slightly loose	Slightly bloody	1–5%
2	Loose	Bloody	6–15%
3	Water diarrhea	Gross bleeding and blood at anus	16–20%
4			>20%

**Table 2 ijms-22-08699-t002:** Histological score to quantify the degree of colitis.

Score	Inflammation	Extend	Regeneration	Crypt Damage	Percent Involvement
0	None	None	-	None	0%
1	Slight	Mucosa	Almost complete regeneration	Basal 1/3 damaged	1–25%
2	Moderate	Mucosa + Submucosa	Regeneration with crypt depletion	Basal 2/3 damaged	26–50%
3	Severe	Transmural	Surface epithelium not intact	Entire crypt and epithelium lost	51–75%
4			No tissue repair		76–100%

## Data Availability

Microbiome sequencing data have been submitted to the NCBI Short Read Archive repository under the BioProject accession number PRJNA694316 (https://www.ncbi.nlm.nih.gov/sra/PRJNA694316).

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
