# Peer review of "Smectite as a Preventive Oral Treatment to Reduce Clinical Symptoms of DSS Induced Colitis in Balb/c Mice"

_ijms, 2021, doi:10.3390/ijms22168699_

Round 1

Reviewer 1 Report

In this paper, Breitrück et al describe that prophylactic treatment with smectite can reduce the symptoms of DSS induced colitis in mice. They show that it reduces weight loss and disease activity index along the course of inflammation and that mice present with lower histological score at sacrifice. They show that this effect does not go through the modulation of claudin expression but might modify the DSS-induced dybiosis.

MAJOR COMMENTS

Overall the study is well done. However, the pertinence of a prophylactic treatment for this kind of disease is not very high. As said by the authors in the discussion, trying to highlight a possible therapeutic effect of smectite would be much more relevant. Author have to do this experiment and compare with the prophylactic treatment. They could also try to see the prophylactic treatment in a model of relapse with multiple cycle of DSS, to see if smectite would ameliorate of even prevent from relapse.

Author also have to add an important control along the study, i.e. the FI5pp-treated group alone, without DSS.

Author could try to test the effect of their compound on the intestinal permeability with FITC-dextran to complete their claudin study.

Reviewer 2 Report

In this reviewed study authors report the effects of in vivo studies on Balb/c mice where smectite reduced clinical symptoms of DSS induced colitis. The findings are based on histology and morphology of large intestine, analysis of selected tight junction proteins and microbiome analysis as well as clinical evaluation of colitis. Although the study seems to be interesting, and results are promising some drawbacks should draw authors attention and be corrected before article publication.

Major comments:

  1. Is it necessary to mention “from Germany” according to diosmectite? Is there something extraordinary in this particular substance? If so, please provide detailed on what differentiate this special source form other sources of smectite.
  2. Provide the number of used animals in the abstract section (lines 33-34)
  3. In the abstract example of crucial results should be presented along with P values.
  4. Please do not use “significant” but P value instead.
  5. The sentence (lines 34-35) does not match results form figure 3.
  6. No clearly stated hypothesis in the introduction.
  7. Avoid using “markedly lower” what does it mean anyway, how much lower is it? Provide proportion or percentage (a P value would also be useful)
  8. Why did the authors not use the repeated measures ANOVA to assess variability in clinical data? Point-by-point (time) comparisons are not fully convincing, and do not show the true effect of the treatment.
  9. Authors must provide greater magnification to show the epithelial damage.
  10. Part of the discussion section, lines 236-251, is not covered by the authors' findings and the work conducted because the authors did not analyze mucins. It is not relevant and should be removed or at least reworded.
  11. Were the mice kept individually in separate cages? If not, how many mice were kept in a cage? There is no information on this throughout the text. This has crucial meaning for the performed statistical analysis. Was the power analysis performed (the n is small – was it enough?). If the small number of animals was used what is the effects size of presented differences? The statistical model described by the authors does not account for variation in repetition. By the way – the authors do not describe or state the statistical model in the relevant Materials and Methods section, and the reader must guess.
  12. Why there is different number of mice in particular groups? There is no explanation. Animal dead during experiment must be reported (DSS and FI5pp + DSS groups?).

Minor comments:

  1. All abbreviation should be explained at the first appearance in the text, also refers to the abstract.
  2. Abbreviation DSS is not explained.
  3. Line 53 – UC instead of CU? CD instead of MC.
  4. Lines 94-98, 131-134, 143, 149-152 belongs to the Materials and Methods section.
  5. Lines 137-140 belongs to the discussion section.
  6. The sentence lines 102-103 make no sense – change “DSS- and prophylactic FI5pp-“ into “Fl5pp + DSS”, it will be consistent with figures.
  7. What fixative was used in the histology procedures?

Round 2

Reviewer 1 Report

I thank the authors for having taken the time to answer my concerns.

I do understand that it will take time to do a chronic DSS, or to test the treatment option and that this part of work will be suitable for another study. The authors justified very well their opinion

However, the absence of the FI5pp-treated group does not allow to draw solid enough conclusion from the work presented. I would encourage the authors to resubmit their manuscript once these controls will be included.

Author Response

Thank you to the reviewer for the helpful comments.

Reviewer 2 Report

I appreciate the authors' responses and have no further comments on this manuscript. In my opinion, it is suitable for publication in its present form. I wish you the best of luck.

Author Response

(The authors gave the same response as above.)
